# TEMPORAL DISENTANGLEMENT OF REPRESENTATIONS FOR IMPROVED GENERALISATION IN REINFORCEMENT LEARNING

**Mhairi Dunion**
University of Edinburgh
mhairi.dunion@ed.ac.uk

**Trevor McInroe**
University of Edinburgh
t.mcinroe@ed.ac.uk

**Kevin Sebastian Luck**
Aalto University
kevin.s.luck@aalto.fi

**Josiah P. Hanna**
University of Wisconsin – Madison
jphanna@cs.wisc.edu

**Stefano V. Albrecht**
University of Edinburgh
s.albrecht@ed.ac.uk

## ABSTRACT

Reinforcement Learning (RL) agents are often unable to generalise well to environment variations in the state space that were not observed during training. This issue is especially problematic for image-based RL, where a change in just one variable, such as the background colour, can change many pixels in the image. The changed pixels can lead to drastic changes in the agent's latent representation of the image, causing the learned policy to fail. To learn more robust representations, we introduce **TE**mporal **D**isentanglement (TED), a self-supervised auxiliary task that leads to disentangled image representations exploiting the sequential nature of RL observations. We find empirically that RL algorithms utilising TED as an auxiliary task adapt more quickly to changes in environment variables with continued training compared to state-of-the-art representation learning methods. Since TED enforces a disentangled structure of the representation, our experiments also show that policies trained with TED generalise better to unseen values of variables irrelevant to the task (e.g. background colour) as well as unseen values of variables that affect the optimal policy (e.g. goal positions).

## 1 INTRODUCTION

Real-world environments are often not static and deterministic, but can be subject to changes, both incremental or with sudden effects (Luck et al., 2017). Reinforcement Learning (RL) algorithms need to be robust to these changes and adapt quickly. Moreover, since many real-world robotics applications rely on images as inputs (Vecerik et al., 2021; Hämäläinen et al., 2019; Chebotar et al., 2019), RL agents need to learn robust representations of images that remain useful after a change in the environment. For example, a simple change in lighting conditions can change the perceived colour of an object, but this should not affect the agent's ability to perform a task.

One of the reasons RL agents fail to generalise to unseen values of environment variables, such as colours and object positions, is that they overfit to variations seen in training (Zhang et al., 2018). The issue is especially problematic for image-based RL, where a change in one environment variable can mean the agent is presented with a very different set of pixels for which trained RL policies are often no longer optimal. This failure to generalise occurs for both variables that are irrelevant to the optimal policy, such as background colours, and relevant variables, such as goal positions (Kirk et al., 2022). In practice, this often results in agents needing to adapt their policy after a change to only one variable. We show experimentally that agents often cannot recover the optimal policy after the environment changes because it is too difficult to 'undo' the overfitting.

One approach to tackle the generalisation issue is to use domain randomisation during training to maximise the environment variations observed (Cobbe et al., 2019; Chebotar et al., 2019). However, in practice, we may be unaware of what variations an agent might see in the future. Even if all possible future variations are known, training on this full set is often sample inefficient and may

result in a sub-optimal policy as the agent learns to compensate for many different values. It can also be impractical to generate all variations during training due to limitations on laboratory setups or in simulation suites. An alternative to domain randomisation is to learn robust representations that generalise to unseen variations (Kirk et al., 2022), but learning such representations remains an open problem (Lan et al., 2022). Some approaches aim to learn a representation that is invariant to image distractors (Zhang et al., 2020; 2021) to improve generalisation to unseen values of only variables irrelevant to the optimal policy. These approaches do not enforce a structure on the remaining task-relevant variables in the representation for generalisation to unseen values of relevant variables.

A promising direction towards robust RL is to learn a disentangled representation of observations. Disentanglement techniques aim to learn robust representations for both task relevant and irrelevant variables by separating distinct, informative factors of variation in an image into the unknown ground truth factors that generated the image (Bengio et al., 2013). When one factor of variation changes to a previously unseen value, such as a new colour, changing many pixels in the image, only a subset of features in a disentangled representation will change. The RL agent will still be able to rely on the remaining unchanged features in the representation to adapt quickly, allowing generalisation and performance recovery similar to state-based RL. Higgins et al. (2017b) show that a disentangled representation improves generalisation in RL. However, this approach requires a trained or hard-coded policy to collect independent and identically distributed (i.i.d.) data to pre-train a $\beta$-VAE (Higgins et al., 2017a) offline, and it has since been proven that it is theoretically impossible to learn a disentangled representation from i.i.d. data alone (Locatello et al., 2019).

We introduce a self-supervised auxiliary task for learning disentangled representations for the robust encoding of images, which we call **TE**mporal **D**isentanglement (TED). Unlike previous work, TED can be implemented with only minimal changes to existing RL algorithms and allows for lifelong learning of disentangled representations. In contrast to Higgins et al. (2017b), our approach uses the non-i.i.d. temporal data from consecutive timesteps in RL to learn the disentangled representation online. Note that TED does not require a decoder which lessens computational costs. We provide experimental results across a variety of tasks from the DeepMind Control Suite (Tunyasuvunakool et al., 2020), Panda Gym (Gallouédec et al., 2021) and Procgen (Cobbe et al., 2020) environments. For our experiments, we train on a subset of some of the environment variables (such as colours), then evaluate generalisation on a test environment with unseen values of the variables and continue training to demonstrate adaptation to the new environment. Our results demonstrate that TED improves generalisation of a variety of base RL algorithms on unseen environment variables that are relevant or irrelevant to the task, while state-of-the-art baselines that achieve equally good training performance still fail to adapt and, in some cases, are unable to recover after overfitting to the training environment. We also evaluate a disentanglement metric (Higgins et al., 2017a) to demonstrate that our approach increases the extent to which the learned representation has disentangled the factors of variation in the image observations compared to baselines.

## 2 RELATED WORK

### 2.1 GENERALISATION IN IMAGE-BASED REINFORCEMENT LEARNING

**Image augmentation.** Image augmentation artificially increases the size of the dataset by adding image perturbations to improve robustness of representations. Laskin et al. (2020a) apply a variety of image augmentation techniques such as translation, cutouts and cropping; Yarats et al. (2021) average over multiple augmentations; Hansen & Wang (2021) maximise mutual information between the representations of augmented and non-augmented images; and Hansen et al. (2021) use both augmented and non-augmented images to stabilise Q-value estimation. However, image augmentation approaches to generalisation can still fail when the agent experiences stronger types of variation after training (Kirk et al., 2022). In our experiments, we show that TED can be used alongside image augmentation techniques to further improve generalisation while benefiting from the augmentation.

**Learning invariant representations.** Invariance techniques aim to learn a representation that ignores distractors in the image. Zhang et al. (2020) use causal inference techniques assuming a block MDP structure; Zhang et al. (2021) use a bisimulation metric; and Li et al. (2021) use domain adversarial optimisation to learn a representation invariant to distractors. These approaches all aim to generalise to unseen values of irrelevant variables, e.g. background colours. In contrast, TED uses disentanglement to enforce a structured representation that applies to *both* relevant and irrelevant variables.

**Encoding inductive biases.** Some approaches use an auxiliary task to encode inductive biases. Laskin et al. (2020b) learn a representation to maximise similarity between different augmentations of the same observation; Mazoure et al. (2020) maximise similarity between observations at successive timesteps; and Agarwal et al. (2021) enforce a structured representation for task relevant variables using policy similarity metrics. These approaches are based on enforcing a similarity constraint between pairs of observations to learn informative features but they do not enforce any structure to the representation, whereas TED encourages a disentangled structure due to the form of the classifier without a similarity constraint. van den Oord et al. (2018) learn representations that capture information predictive of the future, Schwarzer et al. (2021) pretrain an encoder and fine-tune on task specific data, and Jaderberg et al. (2017) uses a combination of multiple auxiliary tasks for representation learning, but these approaches also do not require a disentangled structure. To encode the inductive bias of disentanglement, Higgins et al. (2017b) train a $\beta$-VAE offline using data from a pre-trained or hardcoded agent. In contrast, our approach uses the temporal structure of data available in RL to learn a disentangled representation online with the RL policy.

## 2.2 DISENTANGLED REPRESENTATIONS

**Unsupervised learning.** Many variations of the Variational Autoencoder (VAE) (Kingma & Welling, 2014) aim to improve disentanglement of the learned representation, such as the $\beta$-VAE (Higgins et al., 2017a) (Burgess et al., 2017) and the Factor-VAE (Kim & Mnih, 2018). Locatello et al. (2019) prove that it is theoretically impossible to learn disentangled representations from i.i.d. data alone. To bypass the impossibility result, many recent approaches extend the $\beta$-VAE to use some form of supervision. Shu et al. (2020) provide weak supervision using a labelled grouping of images, while Locatello et al. (2020) generate pairs of images with limited factors of variation between pairs. In contrast, our approach exploits non-i.i.d. temporal observations available in RL to learn a disentangled representation without labelling or artificially generating images.

**Independent component analysis.** Hyvärinen & Morioka (2016) introduce Time-Contrastive Learning to learn a disentangled representation from time-series data with *non-stationary* factors. Hyvärinen & Morioka (2017) disentangle time-series data with *stationary* factors, introducing Permutation Contrastive Learning (PCL) to train a classifier to discriminate between temporal and non-temporal inputs. We consider RL episodes back-to-back as a time-series. Features that are randomised at the start of an episode are stationary, and due to the agent learning and adapting behaviour over time, features controlled by the agent are non-stationary. The TED classification objective is based on the PCL classifier structure to encourage disentanglement, but unlike PCL, TED addresses the combination of stationary and non-stationary features in RL.

## 3 PRELIMINARIES

We assume the environment is a fully-observable Markov Decision Process (MDP), defined as the tuple $\mathcal{M} = (\mathcal{S}, \mathcal{A}, P, R, \gamma)$, where $\mathcal{S}$ is a set of states, $\mathcal{A}$ is a set of actions, $P : \mathcal{S} \times \mathcal{S} \times \mathcal{A} \to [0, 1]$ is the state-transition function, $R : \mathcal{S} \times \mathcal{A} \to \mathbb{R}$ is the reward function, and $\gamma \in [0, 1)$ is the discount factor. An RL agent chooses an action $\mathbf{a}_t \in \mathcal{A}$ at time $t$ based on its current state $\mathbf{s}_t \in \mathcal{S}$ and its policy $\mathbf{a}_t \sim \pi(\mathbf{s}_t)$. The agent then transitions to the next state according to the state-transition probability $P(\mathbf{s}_{t+1}|\mathbf{s}_t, \mathbf{a}_t)$, and receives a reward, $r_t = R(\mathbf{s}_t, \mathbf{a}_t)$. The goal of an RL agent is to learn a policy $\pi$ to maximise the expected discounted cumulative rewards, $\max_\pi \mathbb{E}_{P,\pi}[\sum_{t=0}^\infty [\gamma^t r_t]]$.

In this work, the agent's observation $\mathbf{o}_t \in \mathcal{O}$ at time $t$ is image pixels, a high-dimensional representation of the true underlying environment state $\mathbf{s}_t \in \mathcal{S}$. We assume the environment has a factored state representation $\mathbf{s}_t$, where each factor corresponds to an environment variable. The components of the state vector $\mathbf{s}_t$ are the unobserved ground truth factors of variation. We consider one observation to be a stack of consecutive frames to ensure all features of $\mathbf{s}_t$, such as velocities, can be extracted from $\mathbf{o}_t$. We assume $\mathbf{o}_t$ is generated by an invertible, non-linear transformation of the state factors $\mathbf{s}_t$, $\mathbf{h} : \mathcal{S} \to \mathcal{O}$, such that $\mathbf{o}_t = \mathbf{h}(\mathbf{s}_t)$. The aim of disentanglement is to learn a representation $\mathbf{z}_t \in \mathcal{Z}$ that recovers the independent ground truth factors of $\mathbf{s}_t$ from the observations $\mathbf{o}_t$ by approximating the inverse of the transformation, such that $\mathbf{z}_t = \mathbf{f}(\mathbf{o}_t)$. To simplify notation, we will usually denote $\mathbf{z}_t = \mathbf{f}(\mathbf{o}_t)$ as $\mathbf{z}(\mathbf{o}_t)$. We will use $z^i$ to denote the $i$-th component of the vector $\mathbf{z}$.

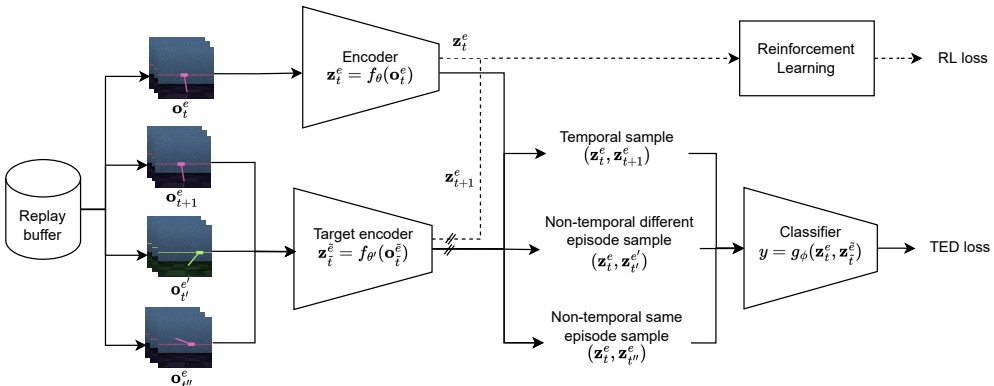

**Figure 1:** TED architecture: The classifier is trained to discriminate between temporal and non-temporal samples to encourage the encoder to disentangle the temporal structure in the image observations. The '//' indicates that the gradient flow is stopped, $\tilde{e} \in \{e, e'\}$ and $\tilde{t} \in \{t+1, t', t''\}$ depending on the observation being processed.

## 4 TEMPORAL DISENTANGLEMENT

We introduce *TEmporal Disentanglement* (TED) as an auxiliary task for representation learning with RL algorithms. Our goal is to improve generalisation to both relevant and irrelevant features when their testing variation is unknown a priori. TED aims to disentangle the factors of variation that generate an observation $\mathbf{o}_t$, such as background colour and the trajectory of an object across the frame stack. We will provide an overview of architecture in Section 4.1 then discuss the details of the TED auxiliary task in Section 4.2.

### 4.1 ARCHITECTURE OVERVIEW

The high-level architecture for TED with a generic RL algorithm is depicted in Figure 1. TED is designed to encourage the encoder $f_\theta : \mathcal{O} \to \mathcal{Z}$ to recover the temporal structure determined by observations at consecutive timesteps, $\mathbf{o}_t$ and $\mathbf{o}_{t+1}$, such that a classifier $g_\phi : \mathcal{Z} \times \mathcal{Z} \to \mathbb{R}$ can discriminate between temporal and non-temporal pairs of observation encodings. We structure the classifier to compare each feature in the representation $\mathbf{z}_t$ separately to encourage the encoder to disentangle the temporal structure into the ground truth factors of $\mathbf{s}_t$ that generated the observation. We simultaneously perform dimensionality reduction such that $\dim(\mathcal{S}) \leq \dim(\mathcal{Z}) \ll \dim(\mathcal{O})$.

TED requires a batch of $N$ transitions $B = \{(o_t, o_{t+1})_i\}_{i=1}^N$ originating from various episodes. The classifier loss is applied as an auxiliary loss to any base RL algorithm. For off-policy algorithms, the batch can be sampled from the replay buffer $B \sim \mathcal{D}$ following the sampling procedure for the base RL algorithm. For on-policy base algorithms, the batch can be created using multiple parallel environments to ensure transitions from different episodes, which is common in practice, for which $B = \mathcal{D}$ in the subsequent explanations. Image augmentations can be applied to the observations where required by the base algorithm, i.e. TED uses the same augmented images as the base algorithm. Both the TED classification loss and the relevant RL loss are used to learn the encoder parameters $\theta$. For training stability, we also use a target encoder (He et al., 2020; Laskin et al., 2020b) $f_{\theta'} : \mathcal{O} \to \mathcal{Z}$ for the next observation at a given timestep $\mathbf{o}_{t+1}$, where $\theta' = \tau\theta' + (1 - \tau)\theta$. Only temporal transitions are used for RL, but non-temporal pairs $(o_t, o_{\tilde{t}})$ are created where $\tilde{t} \neq t+1$ to train the classifier, which we describe in more detail in the next section.

### 4.2 TEMPORAL DISENTANGLEMENT CLASSIFIER

We use the TED auxiliary loss to train a classifier to discriminate between temporal and non-temporal pairs of observation encodings. TED encourages the encoder to learn a representation that uncovers the temporal structure in the data to enable distinguishing whether two observations occurred consecutively or not.

**Classifier inputs.** For each batch of transitions $B \sim \mathcal{D}$, we create three types of observation-pair batches for classifier input: 1) temporal samples $X$, 2) different episode, non-temporal samples $X'$, and 3) same episode, non-temporal samples $X''$. A single transition contains $(\mathbf{o}_t^e, \mathbf{o}_{t+1}^e) \in B$ where $e$ is the episode index corresponding to transition. The temporal sample $\mathbf{x}_t = (\mathbf{o}_t^e, \mathbf{o}_{t+1}^e)$

---

**Algorithm 1** TED update step

---

**Input:** batch of transitions $B = \{..., (\mathbf{o}_t^e, \mathbf{o}_{t+1}^e), ...\} \sim \mathcal{D}$
Create batch $Z_{\text{obs}}$ of representations for each observation in $B$: $\mathbf{z}_t^e = f_\theta(\mathbf{o}_t^e)$
Create batch $Z_{\text{next\_obs}}$ of representations for each next observation in $B$: $\mathbf{z}_{t+1}^e = f_{\theta'}(\mathbf{o}_{t+1}^e)$
**for** each transition in $B$ **do**
    Create temporal sample $\mathbf{x}_t \leftarrow (\mathbf{z}_t^e, \mathbf{z}_{t+1}^e)$, $X \leftarrow X \cup \mathbf{x}_t$
    Sample $\mathbf{z}_{t'}^{e'} \sim Z_{\text{next\_obs}}$ such that $e' \neq e$
    Create non-temporal different episode sample $\mathbf{x}_t' \leftarrow (\mathbf{z}_t^e, \mathbf{z}_{t'}^{e'})$, $X' \leftarrow X' \cup \mathbf{x}_t'$
    Sample $\mathbf{o}_{t''}^e \sim \mathcal{D}$ such that $t'' \notin \{t, t+1\}$, and get representation $\mathbf{z}_{t''}^e = f_{\theta'}(\mathbf{o}_{t''}^e)$
    Create non-temporal same episode sample $\mathbf{x}_t'' \leftarrow (\mathbf{z}_t^e, \mathbf{z}_{t''}^e)$, $X'' \leftarrow X'' \cup \mathbf{x}_t''$
**end for**
**for** each sample $\mathbf{x} \in \{X, X', X''\}$ **do**
    Classifier prediction $y = g_\phi(\mathbf{x})$ (see Equation 1)
    Calculate binary cross-entropy loss $L_{\text{TED}}(\mathbf{x}, l)$ (see Equation 2)
**end for**
Calculate average loss for the batch $L_{\text{TED}} \leftarrow \text{mean}(L_{\text{TED}}(\mathbf{x}))$
Backpropogate loss to update encoder parameters $\theta$ and classifier parameters $\phi$
Update target encoder parameters $\theta' = \tau\theta' + (1 - \tau)\theta$
**Output:** Loss $L_{\text{TED}}$ and updated parameters $\phi$, $\theta$, and $\theta'$

---

consists of two consecutive observations within a given episode $e$. Due to the episodic nature of RL, we use two types of non-temporal samples. The non-temporal sample $\mathbf{x}_t' = (\mathbf{o}_t^e, \mathbf{o}_{t'}^{e'})$ consists of non-consecutive timesteps from different episodes, where $\mathbf{o}_{t'}^{e'} \sim \mathcal{B}$ such that $e' \neq e$. These samples encourage the encoder, $f_\theta$, to learn a representation, $\mathbf{z}(\mathbf{o}_t^e)$, that disentangles episodic features that are chosen randomly at the start of an episode, such as colours, as this is sufficient to distinguish $\mathbf{x}_t'$ from the temporal sample $\mathbf{x}_t$. The second non-temporal sample $\mathbf{x}_t'' = (\mathbf{o}_t^e, \mathbf{o}_{t''}^e)$ consists of non-consecutive timesteps from the same episode where $\mathbf{o}_{t''}^e \sim \mathcal{D}$. These same episode non-temporal samples encourage the encoder to disentangle features that change during the episode, such as agent and object positions, because episodic features will be the same for both $\mathbf{x}_t$ and $\mathbf{x}_t''$.

**Classification objective.** Given a batch of samples $\{X, X', X''\}$, a logistic regression classifier is trained to discriminate between the corresponding representation of temporal samples, $\mathbf{x}_t \in X$, and non-temporal samples, $\mathbf{x}_t' \in X'$ and $\mathbf{x}_t'' \in X''$. To learn a disentangled representation, we use a regression function of the form proposed by Hyvärinen & Morioka (2017):

$$y(\mathbf{x}_t) = \mathbf{g}_\phi(\mathbf{x}_t^1, \mathbf{x}_t^2) = \sum_{i=1}^n |k_1^i z^i(\mathbf{x}_t^1) + k_2^i z^i(\mathbf{x}_t^2) + b^i| - (\bar{k}^i z^i(\mathbf{x}_t^1) + \bar{b}^i)^2 + c \quad (1)$$

for all $\mathbf{x}_t \in \{X, X', X''\}$, where $n$ is the dimensionality of the representation $\mathbf{z}(\mathbf{o}_t^e)$, $\mathbf{x}_t = (\mathbf{x}_t^1, \mathbf{x}_t^2)$, and $\phi = \{\mathbf{k}_1, \mathbf{k}_2, \mathbf{b}, \bar{\mathbf{k}}, \bar{\mathbf{b}}, c\}$ are the classifier parameters to be trained simultaneously with the encoder parameters $\theta$.

Temporal samples, $\mathbf{x}_t \in X$, are given a classification label $l = 1$, and non-temporal samples, $\mathbf{x}_t' \in X'$ and $\mathbf{x}_t'' \in X''$, are given a classification label $l = 0$. The classifier is trained using the cross-entropy loss for binary classification for all $\mathbf{x}_t \in \{X, X', X''\}$:

$$\mathcal{L}_{\text{TED}}(\mathbf{x}_t, l) = -\alpha(2l \log \sigma(y(\mathbf{x}_t)) - (1 - l) \log(1 - \sigma(y(\mathbf{x}_t)))) \quad (2)$$

where $\sigma$ is the sigmoid function. Since there are two non-temporal samples, $\mathbf{x}_t' \in X'$ and $\mathbf{x}_t'' \in X''$, for each transition in the batch $B$ but only one temporal, $\mathbf{x}_t \in X$, the positive temporal samples are weighted by 2. The coefficient $\alpha$ is a hyperparameter to be tuned to the task. It is used to scale up the classification auxiliary loss to ensure it is not dwarfed by the RL loss and to prioritise representation learning over policy learning at the start of training while the factors of variation, $\mathbf{s}_t$, are independent. The structure of the classifier $y = \mathbf{g}_\phi(\mathbf{x}_t^1, \mathbf{x}_t^2)$ (Equation 1) ensures each feature $z^i$ is considered separately to the other features $z^{j \neq i}$. This encourages the encoder to not only uncover the temporal structure necessary for classification, but to do so by separating the factors of variation into distinct features in the representation so that the temporal structure of each feature can be determined independently of the other features. The second term of Equation 1 approximates the marginal log-pdf. Due to this structure, we expect the encoder to learn a disentangled representation and the classifier to approximate the distribution of the samples (Hyvärinen & Morioka, 2017). The

pseudocode for an update step is provided in Algorithm 1, which is performed for every update of the base RL algorithm. We train the encoder with both $\mathcal{L}_{\text{TED}}$ and the RL loss. It is important to note that the TED auxiliary loss requires only creating temporal and non-temporal pairs of representations to train the classifier. The classifier is not required for execution so it can be discarded after training.

# 5 EXPERIMENTAL RESULTS

Our experiments are designed to evaluate whether TED allows zero-shot generalisation to unseen values of an environment variable, and whether TED promotes faster adaptation of the base RL algorithm after a change in environment variables if generalisation is not instant. We evaluate our approach on settings where the agent must generalise to unseen values of *both* task relevant and irrelevant distractor variables with continued learning on the test environment. We use a training environment with a subset of some of the environment variables (such as colours), and evaluate generalisation on a test environment with unseen values of the variables. Images of the train and test environments are provided in Appendix C. We show results on a variety of different tasks with RAD (Laskin et al., 2020a), SVEA (Hansen et al., 2021) and PPO (Schulman et al., 2017) as different base algorithms utilising the TED loss, which covers on-policy and off-policy, continuous and discrete control, and with and without image augmentations, demonstrating the flexibility of our approach. Our results show that TED consistently improves the generalisation of the base RL algorithm in all tasks. TED also shows lower variance across seeds compared to baselines, making it more reliable.

## 5.1 GENERALISATION TO TASK-IRRELEVANT VARIABLES

To demonstrate generalisation to unseen values of task-irrelevant environment variables, we use continuous control tasks from the DeepMind Control Suite (DMC) (Tunyasuvunakool et al., 2020) and Panda Gym (Gallouédec et al., 2021) as simulations of robotics tasks. We adapt DMC wrappers from the Distracting Control Suite (Stone et al., 2021) to add colour distractors to the observations as irrelevant factors of variation.

**Experiment setup.** We show results with RAD and SVEA as two different base algorithms for continuous control. We use the cartpole_swingup and walker_walk tasks from DMC with SVEA as the base algorithm. For RAD, we use the easier finger_spin task instead of walker_walk since RAD was unable to learn an optimal policy on the walker_walk task due to the difficulty of the task with the colour distractors. We also use the Reach task with dense rewards from Panda Gym where the agent receives a reward of the negative of its distance to the goal. We train each algorithm on a fixed set of colours by varying the RGB colour values within a small bounded region of the original value. We test on a set of colours of the same size that were not observed during training. Following the original setup of the base RL algorithms, the encoder for SVEA has 11 convolutional layers and RAD has 4 convolutional layers. Full implementation details are described in Appendix B.

**Baselines.** We compare to the base RL algorithm for each task to demonstrate the performance improvement achieved by TED. For further comparison, we also compare our method to representative baselines from each category of RL generalisation method discussed in Section 2.1. Our data augmentation baseline is DrQ (Yarats et al., 2021). We use DBC (Zhang et al., 2021) as a baseline method that learns invariant representations. We also compare with CURL (Laskin et al., 2020b) as a state-of-the-art contrastive auxiliary task. Finally, we include the base algorithm with domain randomisation, shown as {base algorithm}-DR in the figures, as a privileged baseline that is trained on both the 'train' and 'test' colours together, demonstrating that when the test variations are known a priori, it is often less sample efficient to use domain randomisation and sometimes unable to learn the optimal policy. All baselines use the same size encoder as the TED base algorithm on a given task, so results for the same baseline differ slightly depending on the base algorithm it is being compared to. For all baselines, we tuned hyperparameters by grid search and report the best performing ones.

**Disentanglement metric.** To evaluate the learned representations, we measure the disentanglement using the disentanglement metric proposed by Higgins et al. (2017a). The metric measures disentanglement using pairs of images with one factor of variation fixed to the same value in both images and the other factors randomised. For example, the background colour could be held fixed while all other colours and variables are randomly assigned. We use pairs of frame stacks corresponding to $\mathbf{o}_t$ instead of individual images to allow the policy to extract velocity information.

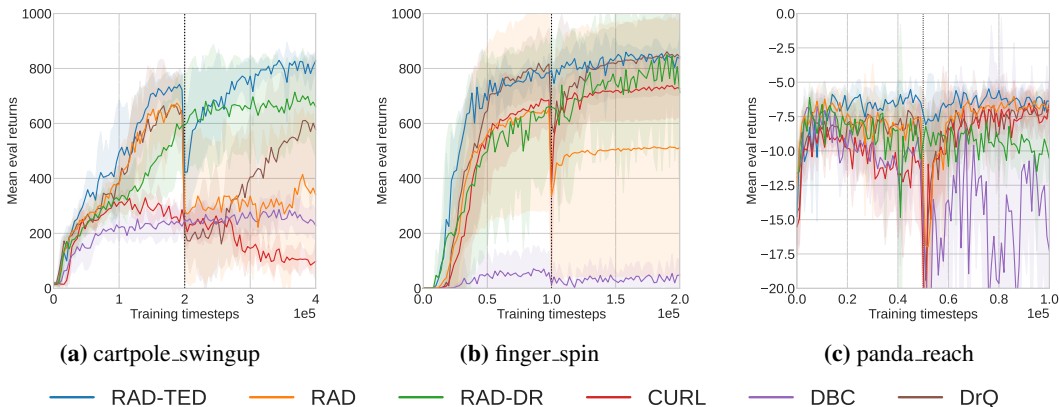

**Figure 2:** Generalisation to unseen colours at the vertical dotted line with RAD base algorithm. RAD-TED (ours) recovers more quickly than baselines and achieves higher returns than domain randomisation (RAD-DR). Returns are the average of 10 evaluation episodes, averaged over 5 seeds, shaded region shows standard deviation.

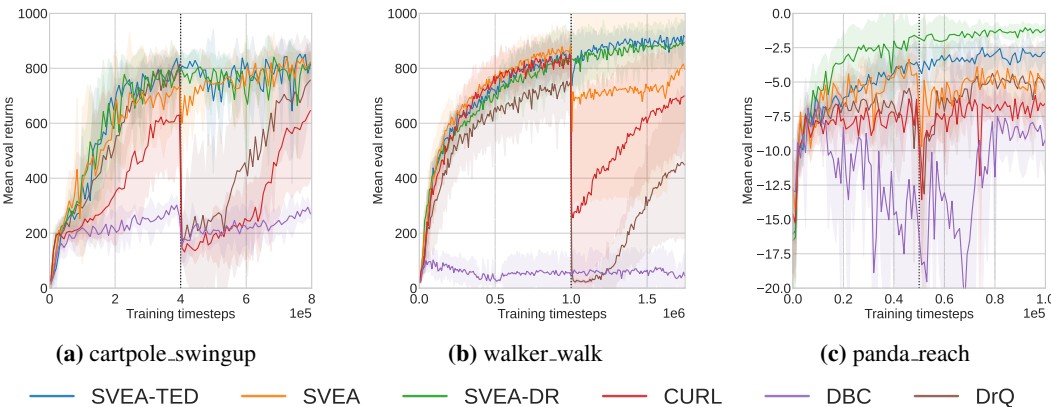

**Figure 3:** Generalisation to unseen colours at the vertical dotted line. SVEA-TED (ours) recovers more quickly than baselines and achieves similar performance to domain randomisation but with fewer assumptions. Returns are the average of 10 evaluation episodes, averaged over 5 seeds, shaded region shows standard deviation.

One sample for the classifier is calculated for a batch $B$ of observation pairs, given by:

$$\mathbf{z}_{\text{diff}} = \sum_{b=1}^{B} |\mathbf{z}(\mathbf{o}_t^{(b)}) - \mathbf{z}(\mathbf{o}_{\tilde{t}}^{(b)})| \tag{3}$$

where $\mathbf{o}_t^{(b)}$ and $\mathbf{o}_{\tilde{t}}^{(b)}$ have the same fixed factor. We then train a linear classifier that takes $\mathbf{z}_{\text{diff}}$ as one input and predicts which factor was held fixed across a batch of inputs. The accuracy of this classifier is the disentanglement score. The intuition is that a there will be low variance in the features corresponding to the fixed factor in a disentangled representation so the classifier will be highly accurate. Although independence of the factors does not strictly hold in practice in many RL environments, it does hold when generating the observations for the metric because the value of each factor is assigned randomly, so we can fairly assess the disentanglement. A score of $1.0$ is the maximum for a fully disentangled representation.

**Results.** The results for RAD as the base algorithm are shown in Figure 2, and the results for SVEA are shown in Figure 3. The figures show that TED improves the generalisation of both RAD and SVEA across all tasks. In many tasks, TED achieves zero-shot generalisation. In other tasks, e.g. Figure 2a, TED experiences some reduction in performance but is able to recover more quickly than the base algorithm and other baselines. Surprisingly, the baselines that achieve equally good training performance as TED fail to adapt and, in some cases, are unable to recover after overfitting to the training colours. Even though TED increases the disentanglement score of the base algorithm, shown in Table 1, there can still be an initial drop in performance on the test environment because the RL policy may still unnecessarily put some weight on the colour features, but the disentangled structure allows faster recovery. TED achieves higher returns than the privileged domain randomisation baseline in many tasks (e.g. Figure 2), which assumes the test colours are known a priori, because

| | cartpole swingup | finger spin | panda reach |
|---|---|---|---|
| RAD-TED | **0.99** | 0.79 | **0.95** |
| RAD | 0.88 | 0.56 | 0.83 |
| RAD-DR | 0.67 | 0.53 | 0.49 |
| CURL | 0.87 | **0.89** | 0.91 |
| DBC | 0.65 | 0.46 | 0.58 |
| DrQ | 0.73 | 0.59 | 0.91 |

| | cartpole swingup | walker walk | panda reach |
|---|---|---|---|
| SVEA-TED | 0.64 | **0.79** | **0.90** |
| SVEA | 0.53 | 0.71 | 0.83 |
| SVEA-DR | 0.58 | 0.62 | 0.72 |
| CURL | **0.92** | 0.77 | 0.65 |
| DBC | 0.63 | 0.61 | 0.34 |
| DrQ | 0.86 | 0.66 | 0.60 |

**(a)** RAD base algorithm (Figure 2) **(b)** SVEA base algorithm (Figure 3)

**Table 1:** Disentanglement scores at the end of training before changing to the test environment.

it can be difficult to learn an optimal policy with such a large set of colour distractors. The poor training performance of some of the baselines, particularly DBC, is due to the difficulty of learning an optimal policy with the colour distractors as it increases the size of the state space. DBC aims to maintain performance on difficult distractors resulting in reduced performance on 'easy' distractors.

## 5.2 GENERALISATION TO BOTH TASK-RELEVANT AND IRRELEVANT VARIABLES

We use the Procgen generalisation benchmark (Cobbe et al., 2020) to demonstrate generalisation to unseen values of *both* task-relevant and irrelevant variables together. Procgen is a set of discrete-control tasks that uses procedural generation to determine the layout, objects, entities, background, colours and other game details for each level of a game. This produces many factors of variation across the game levels, some of which are relevant to solving the game and others are irrelevant distractors. Due to the nature of procedural task generation with Procgen, we are unable to fix the factors of variation to calculate the disentanglement scores for these tasks.

**Experiment setup.** We use the coinrun and jumper Procgen environments. We train on 100 levels of the hard difficulty, and test generalisation on 100 unseen hard levels. Following the setup of Cobbe et al. (2020), we use PPO as the base algorithm for the Procgen tasks.

**Results.** The results are shown in Figure 4. The results show that PPO-TED (ours) recovers more quickly than PPO on the unseen levels. PPO is unable to recover optimal performance on both tasks after overfitting to the training levels, but PPO-TED converges to higher performance on the unseen levels, similiar to that achieved in training. In the jumper environment, Figure 4b, PPO-TED also demonstrates better zero-shot generalisation performance than PPO (at the vertical dotted line).

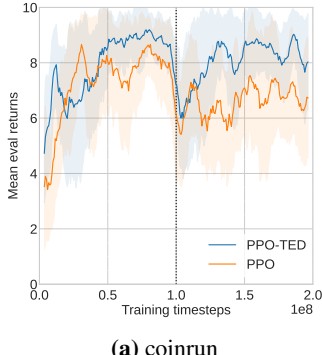
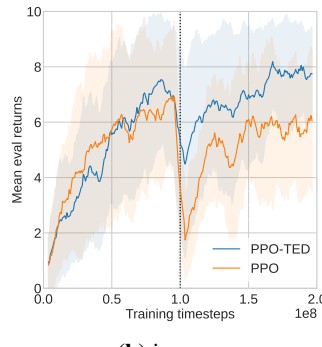

**(a)** coinrun **(b)** jumper

**Figure 4:** Generalisation to unseen levels at the vertical dotted line. PPO-TED (ours) recovers more quickly than PPO on the unseen levels. Returns are the average of 10 evaluation episodes, averaged over 5 seeds, shaded region shows standard deviation. The graphs show the 10-point rolling average for readability.

## 5.3 ABLATION STUDIES

**How does the loss coefficient affect performance?** Figure 5a shows how the choice of the loss coefficient $\alpha$ affects performance. We found $\alpha = 100$ to be optimal for this task, and the results show some robustness as $\alpha = 200$ still achieves good performance. A lower coefficient $\alpha = 50$ does not prioritise disentanglement enough reducing generalisation performance. A higher coefficient reduces performance because the agent is prioritising disentanglement too much over the optimal policy.

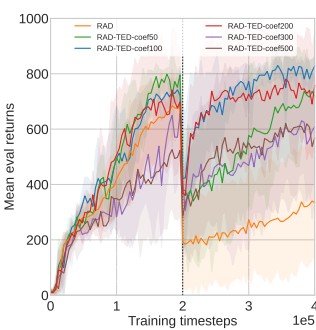 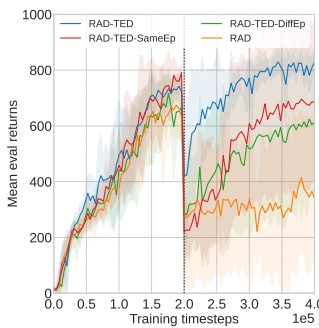 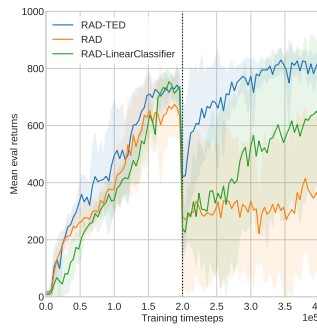

**(a)** Comparison of different TED loss coefficients.

**(b)** TED using same episode or different episode samples only.

**(c)** Modified TED using a standard linear classifier.

**Figure 5:** Ablations for RAD-TED on the cartpole_swingup task with colour distractors, averaged over 5 seeds.

**Is one type of non-temporal sample sufficient to improve generalisation?** TED requires two types of non-temporal samples: from the same episode, and from different episodes. We explored two simplified versions of TED using non-temporal samples from the same episode only and non-temporal samples from different episodes only. The results in Figure 5b show that using a single type of non-temporal sample improves the generalisation of RAD to an extent, but RAD-TED further improves the generalisation performance.

**How important is the disentangled structure of the TED classifier?** We compared TED to a simplified version that uses a standard linear classifier instead of the disentangled structure of the TED classifier. The results in Figure 5c show that while the linear classifier improves the generalisation performance of RAD, it does not generalise as well as RAD-TED.

## 6 LIMITATIONS AND FUTURE WORK

Our approach has some limitations that could be addressed in future work. Guarantees of disentanglement usually assume the factors of variation are independent and do not generalise to correlated factors (Träuble et al., 2021). While we have shown that TED improves disentanglement in practice, RL observations generated by a learning agent do not have independent factors of variation in general. We leave further exploration of how to relax this assumption for future work.

We introduced two types of non-temporal samples to disentangle features controlled by the agent (which form a non-stationary time-series) and episodic features (which form a stationary time-series). While there exists a theoretical proof that disentanglement is possible with either non-stationary (Hyvärinen & Morioka, 2016) or stationary (Hyvärinen & Morioka, 2017) factors, it is still an open problem to identify an approach that guarantees disentanglement in a time-series that contains *both* stationary and non-stationary factors.

Our approach introduces the TED loss coefficient $\alpha$ as a new hyperparameter which must be tuned to the task. Future work could consider automatically tuning $\alpha$ based on the current disentanglement score. Finally, our temporal samples are constructed from the current timestep $t$ and next timestep $t + 1$ to allow TED to be easily be added to existing algorithms. Future work could explore extending the horizon of the temporal sample to $t + k$ with $k > 1$.

## 7 CONCLUSION

In this work, we introduced TED, an auxiliary task for learning disentangled representations in RL. Our approach is the first to consider an auxiliary task based on disentangled representation learning for online RL. TED can be used with existing algorithms and does not require a decoder. We demonstrated experimentally that TED improves generalisation of three different RL base algorithms by adapting with continued learning to previously unseen values of environment variables that are both task-relevant and irrelevant. We also adapted the notion of factors of variation to frame stacking in RL and used a disentanglement metric to show that TED improves the disentanglement of learned representations. TED is a step toward making RL algorithms more robust for real-world deployment and life-long learning as the agent is able to quickly recover when presented with previously unseen values of environment variables and continue learning while reducing catastrophic forgetting.

## ACKNOWLEDGEMENTS

This work was supported by the EPSRC Centre for Doctoral Training in Robotics and Autonomous Systems, funded by the UK Engineering and Physical Sciences Research Council and the Edinburgh Centre for Robotics. This work was also supported by the Academy of Finland Flagship programme: Finnish Center for Artificial Intelligence FCAI. The authors wish to acknowledge the generous computational resources provided by the Aalto Science-IT project and the CSC – IT Center for Science, Finland.

## REPRODUCIBILITY STATEMENT

A public and open-source implementation of TED can be found at github.com/uoe-agents/TED. Full details of the architecture and hyperparameter settings for each of our experiments, including implementation details of the disentanglement metric, can be found in Appendix B.

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

## A    EXTENDED BACKGROUND

In this section, we provide details of the RL algorithms that we use as the base algorithms for TED in our experiments. We use RAD (Laskin et al., 2020a) and SVEA (Hansen et al., 2021) for the experiments in Section 5.1. Both base algorithms are extensions of the Soft Actor-Critic (SAC) algorithm (Haarnoja et al., 2018). We use PPO (Schulman et al., 2017) as the base algorithm in Section 5.2.

**SAC.**    Soft-Actor Critic (SAC) is an off-policy, actor-critic RL algorithm that learns a policy $\pi$ to maximise the expected discounted future rewards and the entropy of the policy. SAC uses transitions from the replay buffer $\mathcal{D}$ to train the critic $Q : \mathcal{O} \times \mathcal{A} \to \mathbb{R}$ by minimising the loss:

$$L_Q = \mathbb{E}_{(\mathbf{o}_t, \mathbf{a}_t, \mathbf{o}_{t+1}, r_t) \sim \mathcal{D}} \left[ \left( Q(\mathbf{o}_t, \mathbf{a}_t) - r_t - \gamma \bar{V}(\mathbf{o}_{t+1}) \right)^2 \right] \tag{4}$$

The target value function $\bar{V}$ is estimated by:

$$\bar{V}(\mathbf{o}_{t+1}) = \mathbb{E}_{\mathbf{a}_{t+1} \sim \pi} \left[ \min_{i=1,2} \bar{Q}_i(\mathbf{o}_{t+1}, \pi(\mathbf{o}_{t+1})) - \alpha_{\text{SAC}} \log \pi(\mathbf{a}_{t+1} | \mathbf{o}_{t+1}) \right] \tag{5}$$

where $\bar{Q}$ is the target Q network whose parameters are an exponential moving average of the corresponding Q network parameters. We maintain two Q networks, $Q_1$ and $Q_2$, and use the minimum of the two networks for the updates. The actor $\pi$ is trained by minimising the loss:

$$L_\pi = \mathbb{E}_{\mathbf{o}_t \sim \mathcal{D}} \left[ \mathbb{E}_{\mathbf{a}_t \sim \pi} \left[ \alpha_{\text{SAC}} \log(\pi(\mathbf{a}_t | \mathbf{o}_t)) - \min_{i=1,2} \bar{Q}_i(\mathbf{o}_t, \mathbf{a}_t) \right] \right] \tag{6}$$

**RAD.**    RAD adds data augmentations to the observations before the SAC network updates. We use image padding and random crop augmentations for the observations in each transition sampled from the replay buffer $(\mathbf{o}_t, \mathbf{o}_{t+1}) \sim \mathcal{D}$.

**SVEA.**    SVEA stabilises Q-learning using a combination of augmented and unaugmented images with the updated loss:

$$L_Q^{\text{SVEA}} = \alpha_{\text{SVEA}} L_Q(o_t, a_t, o_{t+1}) + \beta_{\text{SVEA}} L_Q(o_t^{\text{aug}}, a_t, o_{t+1}) \tag{7}$$

The policy $\pi$ is trained using only unaugmented images, using the standard loss in Equation 6.

**PPO.**    Proximal Policy Optimisation (PPO) is an on-policy, actor-critic RL algorithm for continuous or discrete action spaces. We use a discrete policy $\pi_\psi$ for the Procgen environments. PPO learns a policy by minimising a clipped loss over minibatches of transitions:

$$L_\pi^{\text{CLIP}} = -\mathbb{E}_{(\mathbf{o}_t, \mathbf{a}_t, \mathbf{o}_{t+1}, r_t) \sim \pi} \left[ \min(\rho_t A_t, \text{clip}(\rho_t(\psi), 1 - \epsilon 1 + \epsilon) A_t) \right] \tag{8}$$

where $\rho_t(\psi)$ is the ratio of the action probability under the new and old policies, and $A_t$ is the action advantage given by: $A_t = Q^\pi(o_t, a_t) - V^\pi(o_t)$.

## B    IMPLEMENTATION DETAILS

In this section, we provide the implementation details for TED. Our codebase is built on top of the publicly released DrQ PyTorch implementation by Yarats et al. (2021), and uses the official implementation of SVEA (Hansen et al., 2021). We adapt the codebase to implement the base RL algorithms as well as the TED auxiliary task. A public and open-source implementation of TED is available at github.com/uoe-agents/TED.

### B.1    RAD AND SVEA IMPLEMENTATION DETAILS

**Encoder architecture.**    We use the same encoder architecture as Yarats et al. (2021). The encoder weights are shared between the actor $\pi$ and critic $Q$. For the RAD experiments, the encoder consists of 4 convolutional layers following the original RAD implementation (Laskin et al., 2020a). For SVEA experiments, we use 11 convolution layers following the original SVEA paper (Hansen et al.,

2021). Baselines follow the same encoder size as the base RL algorithm for TED in each experiment. Each convolutional layer has a $3 \times 3$ kernel size and 32 channels. The first layer has a stride of 2, all other layers have a stride of 1. There is a ReLU activation between each convolutional layer. The convolutional layers are followed by a trunk network with a linear layer, layer normalisation, and finally a tanh activation.

Both RAD and SVEA use the critic loss $\mathcal{L}_Q$ to update the encoder parameters. So our implementation of TED requires both the critic loss $\mathcal{L}_Q$ and TED loss $\mathcal{L}_{\text{TED}}$ to update the encoder together. In practice, this can be done by adding the two losses $\mathcal{L}_{\text{ENC}} = \mathcal{L}_Q + \mathcal{L}_{\text{TED}}$ and backpropagating the encoder loss $\mathcal{L}_{\text{ENC}}$ to update the encoder parameters.

**Actor and critic architecture.** The actor $\pi$ and critic $Q$ networks are both 2-layer MLPs with a hidden dimension of 1024. We apply ReLU activations after each layer except the last layer.

**TED architecture.** The TED classifier is implemented with the following parameters: $\mathbf{k_1}$, $\mathbf{k_2}$, $\bar{\mathbf{k}}$, $\mathbf{b}$, and $\bar{\mathbf{b}}$ are vectors of the same size as the latent representation; and $c$ is a scalar. The output of the classifier is defined in Equation 1.

**Hyperparameters.** Table 2 shows the value of the TED loss coefficient $\alpha$ for each task. All other hyperparameters are provided in Table 3. Unless specified otherwise, we use the same hyperparameter settings for both RAD-TED and SVEA-TED algorithms.

| Environment | Base algorithm | Value of $\alpha$ |
|---|---|---|
| cartpole_swingup | RAD | 100 |
| cartpole_swingup | SVEA | 0.5 |
| finger_spin | RAD | 25 |
| walker_walk | SVEA | 1 |
| panda_reach | RAD | 200 |
| panda_reach | SVEA | 0.5 |

**Table 2:** TED loss coefficient $\alpha$.

| Hyperparameter name | Value |
|---|---|
| Replay buffer capacity | 100000 |
| Initial steps | 1000 |
| Stacked frames | 3 |
| Action repeat | 2 for finger_spin, 4 otherwise |
| Batch size | 128 |
| Discount factor | 0.99 |
| Optimizer | Adam |
| Learning rate (actor, critic and encoder) | 1e-3 |
| Target soft-update rate $\tau$ | 0.01 |
| Actor update frequency | 2 |
| Actor log stddev bounds | $[-10, 2]$ |
| Latent representation dimension | 50 for RAD, 100 for SVEA |
| Image size | (84, 84) |
| Image pad | 4 |
| Initial temperature | 0.1 |

**Table 3:** Hyperparameter values for both RAD-TED and SVEA-TED.

**Disentanglement metric.** To calculate the disentanglement metric described in Section 5.1, we collect a batch of 10,000 samples. To create each sample, we use 32 pairs of observations with the same fixed factor to evaluate Equation 3 with $B = 32$. The batch of samples is split into 8,000 training samples to train the classifier, and 2,000 testing samples to calculate the accuracy of the classifier as the disentanglement score. We use a Scikit-learn logistic regression classifier with L1 regularisation and the saga solver.

## B.2 PPO Implementation Details

We follow the PPO architecture and hyperparameters used in the Procgen benchmark (Cobbe et al., 2020). The encoder uses the IMPALA (Espeholt et al., 2018) architecture. PPO is augmented with TED by adding the loss terms $\mathcal{L} = \mathcal{L}_{\text{PPO}} + \mathcal{L}_{\text{TED}}$ and backpropagating the total loss $\mathcal{L}$ with a shared optimiser. The hyperparameters are shown in Table 4.

| Hyperparameter name | Value |
|---|---|
| Image size | (64, 64) |
| Discount factor $\gamma$ | 0.999 |
| GAE $\lambda$ | 0.95 |
| # Timesteps per rollout | 250 |
| Epochs per rollout | 3 |
| # Minibatches per rollout | 8 |
| Entropy bonus | 0.01 |
| PPO clip range | 0.2 |
| Learning rate | 5e-4 |
| # Workers | 1 |
| # Environments per worker | 64 |
| LSTM? | No |
| Frame stack? | No |
| TED coefficient $\alpha$ | 1 for coinrun, 0.25 for jumper |

**Table 4:** Hyperparameter values for PPO-TED

## C Environment Images

In Figure 6, we provide images of example observations for each of the DMC and Panda Gym training and testing environments used in our experiments to visualise the generalisation challenge.

In Figure 7, we provide images of example observations for each of the Procgen environments used in our experiments.

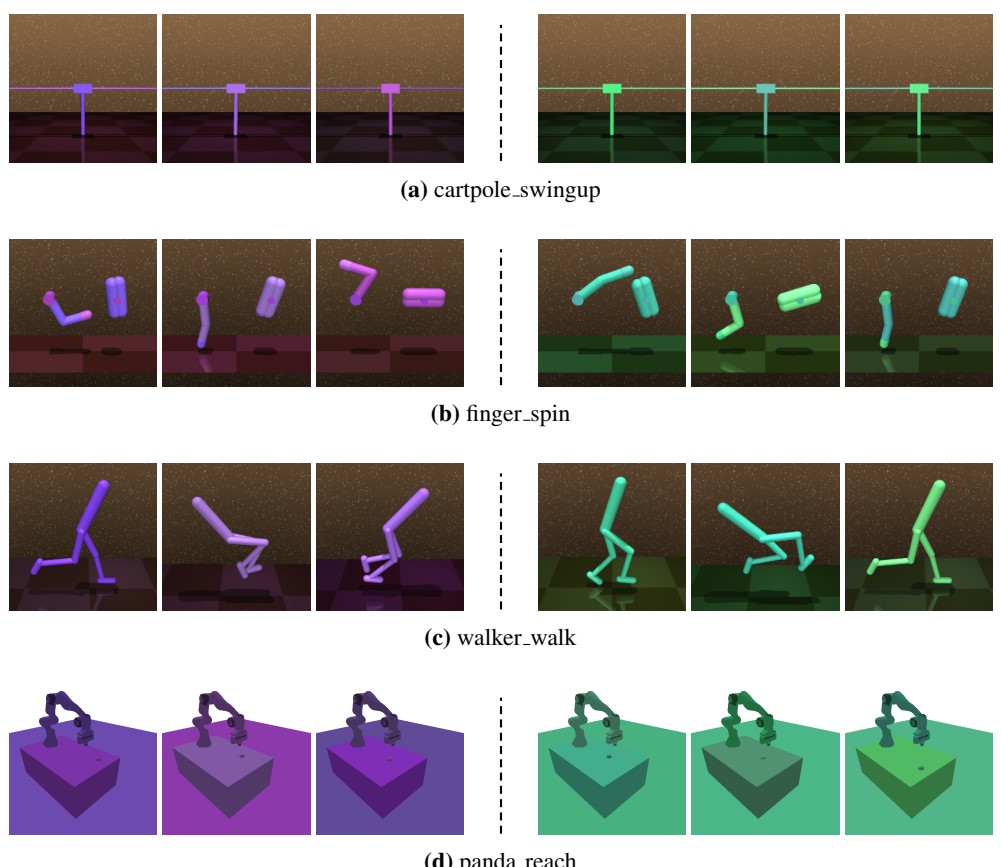

**(a)** cartpole_swingup

**(b)** finger_spin

**(c)** walker_walk

**(d)** panda_reach

**Figure 6:** Example observations for each task with colour distractors used in our experiments (before image pre-processing to reduce the size). The images on the left are examples from the training environment, and the images on the right are examples from the testing environment.

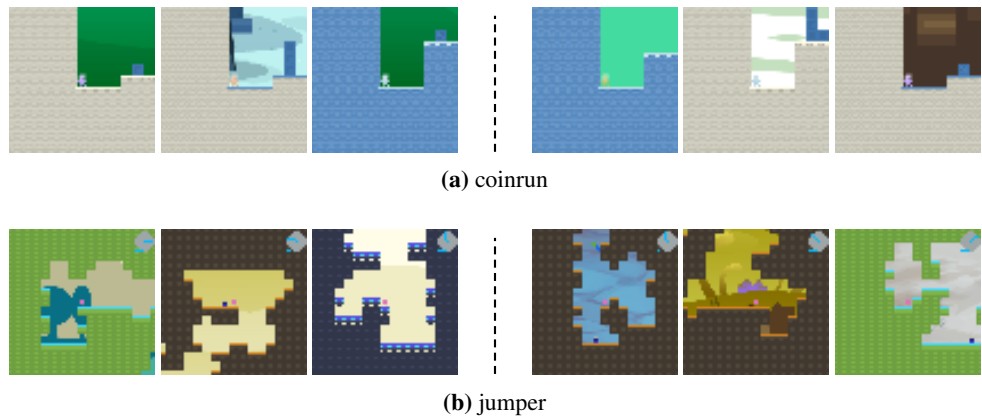

**(a)** coinrun

**(b)** jumper

**Figure 7:** Example observations for each Procgen environment used in our experiments. The images on the left are examples from the training environment, and the images on the right are examples from the testing environment.

