# OpenReview forum: "Temporal Disentanglement of Representations for Improved Generalisation in Reinforcement Learning"
_ICLR.cc/2023/Conference — ICLR 2023 poster_

### Official Review · Reviewer_aCmj · 2022-10-19

**Confidence:** 4
**Correctness:** 3
**Technical Novelty And Significance:** 3
**Empirical Novelty And Significance:** 2
**Recommendation:** 6

**Clarity, Quality, Novelty And Reproducibility:**

The proposed TED auxiliary loss is novel for solving RL problems. The contributions are presented clearly and the code is also provided.

**Strength And Weaknesses:**

Strength:
1. Generalization of RL algorithms is an important research area, and this paper proposed a self-supervised auxiliary loss to achieve improved generalization in RL.
2. The idea of combining the TED auxiliary loss with RL is novel.
3. The paper is clearly presented and well-written.
4. The proposed method is easy to implement and the experiments are conducted on different benchmarks.

Weakness:
1. TED requires non-temporal samples from both the same episode and different episodes. However, the necessity of using non-temporal samples from the same episode is not convincing. Why not just use samples from different episodes or mix the two sources during sampling? The ablation study also does not include experiments that use non-temporal samples from different episodes only.
2. The parameter alpha seems to be a sensitive parameter. However, experiments are only implemented on the cartpole swingup task with three different values. More analysis should be added. Is the optimal value of alpha different in different tasks?
3. In many experiments, the training does not seem to have converged yet. More training steps are suggested. Also, the number of DMC tasks are quite limited in this paper.
4. In Figure 4, what if the TED loss is combined with CURL in the cartpole swingup task.

**Summary Of The Paper:**

The paper proposes to improve the generalization of RL through a self-supervised auxiliary task which aims to learn disentangled representations. The auxiliary loss is used to train a classifier to discriminate between temporal and non-temporal pairs so that the encoder is encouraged to disentangle temporal structure in the observations. The proposed auxiliary loss can be easily combined to existing RL algorithms and experimental results on three benchmarks are reported.

**Summary Of The Review:**

Generalization of RL algorithms is an important topic and the motivation of this paper is good. To solve the generalization problem of RL, the paper proposes a novel disentangled representation learning auxiliary task and experiments on three benchmarks demonstrate its effectiveness. The major concern is the completeness and quality of the experiments.

---

> ### Author Response · Authors · 2022-11-17
> **Response to Reviewer aCmj**
>
> We thank you for your review and insightful comments. We are glad you appreciate the importance of the research area, clarity of the paper, and the novel approach we propose. We would like to address the weakness you raised in detail below (using the same numbering as your review). We have also uploaded an updated version of the paper with several improvements based on the feedback we received; changes are highlighted in blue for your convenience.
>
> > “1. TED requires non-temporal samples from both the same episode and different episodes. However, the necessity of using non-temporal samples from the same episode is not convincing.”
>
> To demonstrate the need for these two types of non-temporal samples, we have amended the ablation study that compares TED with a version using same episode non-temporal samples only, TED-SameEp, to also compare with a version using different episode non-temporal samples only, TED-DiffEp, in Figure 5b of the updated version of the paper. We find that using either same episode or different episode samples improves the generalisation to an extent, but using both types of non-temporal samples together improves the generalisation further.
>
>
>
> > “2. The parameter alpha seems to be a sensitive parameter. However, experiments are only implemented on the cartpole swingup task with three different values.”
>
> We have extended the loss coefficient ablation study (in Figure 5a) to include additional values to provide additional insight. The optimal value of alpha does depend on the task and base algorithm as it must be scaled to be a similar size as the other losses. Hyperparameter values for all experiments are provided in Appendix B.
>
>
>
> > “3. In many experiments, the training does not seem to have converged yet. More training steps are suggested. Also, the number of DMC tasks are quite limited in this paper.”
>
> We continue training on the test environment for the same number of timesteps as the training environment to demonstrate that many baselines are re-learning from scratch and often still can’t recover the training performance after an equal number of timesteps on the test environment due to the overfitting. We chose the tasks from DMC in which SOTA base algorithms could learn an optimal policy with the colour distractors. Our goal is to test generalisation of the optimal policy (or very close to optimal) learned in the training environment. Many of the more difficult DMC tasks mean that SOTA algorithms are unable to learn the optimal policy from images with distractors, which makes it difficult to assess generalisation. Our results show that still many baselines fail to learn an optimal policy in some of the tasks. Our results also show that the baselines fail to generalise in our experiments, demonstrating that the colour deviations are already sufficiently difficult to test generalisation performance. We also include the Panda task as a more realistic robotics task, as well as Procgen which has many more factors of variation beyond colour.
>
>
>
> > “4. In Figure 4, what if the TED loss is combined with CURL in the cartpole swingup task.”
>
> We show results for the TED auxiliary task applied to three different algorithms in the paper. With limited space and computational resources, we are not able to provide results for all baselines with TED applied but hope that the three different base algorithms demonstrate the flexibility of the approach. We would expect to see TED improve the generalisation performance of CURL as well.

---

### Official Review · Reviewer_QVh3 · 2022-10-24

**Confidence:** 4
**Correctness:** 3
**Technical Novelty And Significance:** 3
**Empirical Novelty And Significance:** 3
**Recommendation:** 6

**Clarity, Quality, Novelty And Reproducibility:**

## Novelty and Placement

The idea is novel to the best of my knowledge, but it bears a lot of similarity with [CURL](https://arxiv.org/pdf/2004.04136.pdf). I believe comparing TED to CURL would better emphasize the novel components in TED.

## Support

The claims are mostly adequately supported, but some changes in the training regime can help the comparison to previous work.

Moreover, the paper argues that disentangled representations should help with generalization, and I think it would be useful to have a discussion on this specific point. For example, consider a study where you ablate the classification loss with a more common not-disentangled predictor. The ablation on the loss weight is not too informative here because it is ablating the whole auxiliary task, not really disentanglement.

I am somewhat concerned about the generality of the conclusions about generalization, given the "separation by color" between training and test tasks. However, maybe even this setting is a good starting point, since it's enough to expose some degradation in performance of the baselines.

## Clarity

The paper is clearly written but some improvements can be made:
* clarify the classification objective
* highlight Strengths 1 and 2
* clarify the connections to negative results (related to stationarity and nonstationarity)

## Reproducibility

The paper explains the idea, evaluation and implementation well enough to reproduce. The bands in the plots could be improved from 1 standard deviation to confidence bands to help understand what to expect from similar runs.

**Strength And Weaknesses:**

## Strengths

1. To me, one of the most interesting findings in the paper is that TED improves how fast the agent can adapt to new tasks, and how different the ability to adapt can be for auxiliary tasks that look equally good in terms of training performance.
2. The design that incorporates disentanglement into the predictor is a core algorithmic strength.
3. The empirical study in the paper has wide coverage in terms of algorithms and task suites.
4. The evidence provided adequately supports the claims being made.

## Weakness

1. The temporal sampling is a limitation of this particular method. How limiting is a matter of opinion, but in my opinion it is a strong limitation.
2. In my opinion the focus on demonstrating that TED outperforms previous baselines limits the impact of the paper.
3. Some choices in the empirical study make the results harder to immediately compare to previous work.

**Summary Of The Paper:**

Thank you for your submission. I mentioned the main points in each section, and I elaborated on the points in the Detailed Comments.

## Summary

The paper proposes TED, an auxiliary task for deep RL agents that encourages learning disentangled representations. TED is shown to improve generalization across color variations of training domains, and is evaluated across different task suites and in combination with different "base" RL agents.

**Summary Of The Review:**

## Concerns

My main concern is the impact/relevance of the results for the broader community. Modulo any concerns raised by other reviewers, I will raise my score to accept (8) if:
* the authors improve the clarity of the paper (the three points above)
* can demonstrate that, in the setting considered, TED's disentanglement component makes a big contribution to generalization.

## Detailed Comments

**Strength 1**. I would like the paper to emphasize this a bit more. It helps make a stronger argument for why we should care about the evaluation setting being considered in the paper. Plus, I think this is quite a surprising fact, because we should care about generalization to tasks that are slightly outside our training set, but we rarely do, and as researchers we may inadvertently discarding/deprioritizing research ideas that fail to outperform existing baselines in training, whereas they could be better than the baselines in ways that training performance does not measure, but that we could also care about (i.e., in this paper, adaptation to new tasks).

**Strength 2**. This could also be investigated further. There's a chance that conclusions about disentangled representations generalizing better carry over to other settings, so the conclusions in this work can be useful for other work too.

**Weakness 1**. This is my only concern about the algorithmic contribution. The method is highly dependent on the negative-example distribution "making sense" or making the classification problem challenging enough to drive representation learning. A failure case would be if the time index $t$ can easily be inferred from $o_t$. In that case a representation that discards everything else and retains only information about `t` can classify optimally. This can therefore be an issue with recurrent representations that retain some amount of information about `t` in them. TED may not immediately carry over to settings where we can't design a suitable negative example distribution, but some empirical conclusions of this paper can still be useful more broadly (namely the conclusions from Strengths 1 and 2).

**Weakness 2**. This is related to my request for more emphasis to Strengths 1 and 2, and it's about making explicit some of the conclusions that go beyond the settings where Weakness 2 is not a problem.

**Strength 4 and Weakness 3**. The choice of evaluation (train for N steps, then change tasks and train for N more steps). This is quite different from many previous works on generalization, in particular, previous work that has discussed zero-shot generalization (ZSG). I think this paper made the right decision to look at adaptation because ZSG would not give a full picture of TED's usefulness. However, certain changes could have helped compare the results against the ZSG in previous work. For example, reporting ZSG performances in control tasks at 100k and 500k.

**Comparison to CURL**. I recommend comparing TED to CURL in the presentation because many components are similar. The comparison will help emphasize the main impactful idea in TED, which I believe to be the classification loss that encourages disentanglement. In my opinion the choice of negative example distribution makes sense and is adequate, but is domain dependent (Weakness 1).

**Investigating good ZSG cases**. I noticed that in Figure 3 SVEA-TED is generalizing quite well _zero-shot_. It seems essentially unaffected by the change at the vertical dotted line, and continues to improve the agent in some cases. If this is not an artifact of smoothing, I think it can be quite an interesting finding. For future work, I suggest the authors reflect and investigate what it is about the SVEA+domain combination that is causing this effect, and whether it can be incorporated into the auxiliary task to bear in other domains. Figure 3(b) is a really interesting example of this phenomenon, where the baselines degrade a lot at 1e6 steps, but SVEA-TED and SVEA-DR continue to improve "almost as though" there has been no change in the training distribution.

**Relation to theoretical results**. The paper draws some connections to negative results about disentanglement, but only in Sections 1, 2.2 and 6. Without additional context or specific knowledge of the results, the comments in this paper are vague to the reader. Moreover, the "decomposition" in terms of stationary and non-stationary factors does not seem to have any bearing in the discussion of the results. It would be good to see this clarified, especially if the authors have intuition on the subject that is guiding how they designed or interpret TED.

**Related work**. It may be worth citing [CURL](https://arxiv.org/pdf/2004.04136.pdf) or [MoCo](https://openaccess.thecvf.com/content_CVPR_2020/papers/He_Momentum_Contrast_for_Unsupervised_Visual_Representation_Learning_CVPR_2020_paper.pdf) in reference to the target encoder.
There are some other papers with auxiliary tasks that can be relevant to current and future work, for example: [UNREAL](https://arxiv.org/pdf/1611.05397.pdf), [SimCore](https://proceedings.neurips.cc/paper/2019/file/2c048d74b3410237704eb7f93a10c9d7-Paper.pdf), [PBL](http://proceedings.mlr.press/v119/guo20g/guo20g.pdf) and [SPR](https://arxiv.org/pdf/2007.05929.pdf).

**Figures**. I suggest placing the figures slightly below where they are first mentioned. For example, Figure 2 is a page before the results discussion. Also, should "Figure 4" be "Table 4"? Finally, the curve colors used may be difficult for colorblind readers to interpret. Please consider whether the color scheme in use is accessible or if it can be improved.

**Hyperparameters** While selecting $\alpha$ will often require domain-specific experimentation, I have found that a good rule of thumb for auxiliary tasks with RL is to make the scaled auxiliary loss of the same order of magnitude as the RL loss.

---

> ### Author Response · Authors · 2022-11-17
> **Response to reviewer QVh3 (1 of 2)**
>
> We thank you for your insightful feedback. We are glad you find the TED approach interesting, particularly our experimental results showing improved adaptation with a wide coverage in terms of tasks and algorithms used. We would like address the specific weaknesses you raised in more detail below. We have also uploaded an updated version of the paper with several improvements based on the feedback we received; changes are highlighted in blue for your convenience.
>
> ### Weaknesses
>
> Weakness 1:
> > “The temporal sampling is a limitation of this particular method” and “The method is highly dependent on the negative-example distribution "making sense" or making the classification problem challenging enough to drive representation learning.”
>
> We use non-temporal samples from the same episode and from a different episode to ensure the representation learning task is challenging without being able to rely on just one feature. However, we agree that an environment where the time index can easily be inferred from the representation would pose an issue. The representation would still not discard everything else because we also allow the critic loss to update the encoder, but it may be possible for the classifier to learn with only this one feature disentangled from all other features in the scenario you describe. However, we would argue that this scenario may be less common and less realistic than the environments we considered. As you noted, we believe that the contributions of our paper are still useful in many scenarios, and that this scenario may be an interesting extension for future work.
>
>
>
> Weakness 2:
> > “In my opinion the focus on demonstrating that TED outperforms previous baselines limits the impact of the paper.”
>
> Our main focus is to demonstrate how TED improves the performance of a variety of base RL algorithms and we hope this is clear in the paper. However, we do use additional baselines to put the performance into context and make it easier for readers to see what current SOTA performance is. We have added some additional wording to emphasize this further in the updated version of the paper.
>
>
>
> Weakness 3:
> > “Some choices in the empirical study make the results harder to immediately compare to previous work.”
>
> Our experiments are designed with a focus on adaptation to the unseen test environment, while many SOTA algorithms use a different setup to focus on zero-shot generalisation. Our experiment setting provides a realistic scenario as we can expect to see variation in images during training in real-life, such as lighting changes during training. This experimental setting allows our results to highlight an often overlooked failure in many SOTA algorithms, which is shown through the baseline results we provide. As you point out in your detailed explanation of this comment, SVEA-TED achieves good zero-shot generalisation performance. Indeed this is not an artifact of smoothing, since these graphs do not use smoothing. We believe the data augmentation technique of SVEA also helps provide more stability to the TED auxiliary task which uses the same data augmentations. This results in a relatively robust algorithm, and we agree that further investigation of this successful combination would make interesting future work.
>
>
>
> ### Novelty
>
> > “The idea is novel to the best of my knowledge, but it bears a lot of similarity with CURL. I believe comparing TED to CURL would better emphasize the novel components in TED.”
>
> We have extended Section 2.1 with a slightly more detailed explanation of the difference between TED and CURL (and other contrastive approaches) in the updated version of the paper to make sure it is clear. CURL enforces a similarity between observations but does not use the temporal data and does not enforce any structure to the representation. TED, on the hand, focusses on the disentangled structure without enforcing similarity. We would also like to point out that our experimental results also show that TED achieves better generalisation performance than CURL, as well as better performance on the training environment in many cases.
>
>
>
> ### Support
>
> > "The paper argues that disentangled representations should help with generalization, and I think it would be useful to have a discussion on this specific point. For example, consider a study where you ablate the classification loss with a more common not-disentangled predictor.”
>
> Thank you for this recommendation to make the benefits of the TED classification loss more clear. We have added an additional ablation study to Section 5.3 that compares TED with a simplified variation that uses a standard linear classifier. The results show that while the standard linear classifier improves the generalisation of the base algorithm to an extent, the generalisation is further improved by TED.

---

> ### Author Response · Authors · 2022-11-17
> **Response to Reviewer QVh3 (2 of 2)**
>
> > “I am somewhat concerned about the generality of the conclusions about generalization, given the "separation by color" between training and test tasks. However, maybe even this setting is a good starting point, since it's enough to expose some degradation in performance of the baselines.”
>
> We chose the relatively simple setting of colour distractors in DMC to ensure a setting where SOTA base algorithms could learn an optimal policy. Our goal is to test generalisation of the optimal policy (or very close to optimal) learned in the training environment. Many of the more difficult distractors mean that SOTA algorithms are unable to learn the optimal policy from images, which makes it difficult to assess generalisation. Our results show that still many baselines fail to learn an optimal policy in the training environment in some of the tasks. Our results also show that the baselines fail to generalise in our experiments, demonstrating that the colour deviations are already sufficiently difficult to test generalisation performance. We also use Procgen environments in our experiments which have many more factors of variation beyond colour, including factors that affect the optimal policy.
>
>
> ### Clarity
>
> > "clarify the classification objective”
>
> We have added further discussion to explain the difference to CURL as mentioned above. We would be happy to discuss further if there are other parts of the classification objective that require clarification.
>
>
> > “highlight Strengths 1 and 2”
>
> Strength 1: We are glad you appreciate the focus on adaptability in our experiments. We have added some additional wording to the Introduction and Experimental Results to emphasise this further in the updated version of the paper.
>
> Strength 2: As mentioned above, we have also included an additional ablation study in the updated paper to demonstrate the importance of the disentangled structure of the classifier.
>
>
> > “clarify the connections to negative results (related to stationarity and nonstationarity)”
>
> We highlighted the stationary and nonstationary time-series in Section 2.2 as is it is an important distinguishing feature of two related ICA papers. This terminology is used again in Section 6 when specifically referring back to these works. Neither stationary nor nonstationary ICA approaches can be directly applied to RL since RL contains both types of data, and we provide some intuition behind this reasoning in Section 2.2. The stationary features are the features that are randomised at the start of the episode and the non-stationary features are the features that change during the episode. This reasoning is then carried through to Section 4, where we explain the two types of non-temporal samples to disentangle these two types of features, using the RL terminology rather than the ICA terminology. To demonstrate the need for these two types of non-temporal samples, we have amended the ablation study that compares TED with a version using same episode non-temporal samples only to also compare with a version using different episode non-temporal samples only in the updated version of the paper.
>
>
> ### Detailed Comments (where not already addressed above)
>
> > “Related work”
>
> We have added a citation to both CURL and MoCo in reference to the target encoder in the updated version. We have also added UNREAL to the Related Literature Section 2.1. Thank you for also providing additional references relating to model-based RL which we will indeed consider for future work in this direction.
>
>
> > “the curve colors used may be difficult for colorblind readers to interpret”
>
> Thank you for pointing out potential accessibility problems with the colours. We use the “colorblind-friendly” colour cycle provided by matplotlib but we have changed the opacity in the updated version of the paper to hopefully improve accessibility.

---

### Official Review · Reviewer_rUnk · 2022-10-24

**Confidence:** 4
**Correctness:** 3
**Technical Novelty And Significance:** 3
**Empirical Novelty And Significance:** 3
**Recommendation:** 6

**Clarity, Quality, Novelty And Reproducibility:**

## Clarity

In Section 4.2 it's not clear how the stacks of image observations differ between timesteps. Is it that just a single frame is replaced? This seems pretty limiting in terms of assessing the temporal correlation between observations since a significant amount of information is the same between the observations... How is the encoder prevented from mode collapse? If this is the desired component of disentangled representation learning, wouldn't it be easier to just frame difference the observations and learn from that input representation? I fear that the approach is too focused on a complex solution to perhaps consider more simple alternatives that provide the same benefit.

As discussed above, the overall clarity of the paper is good. I think that Sections 1-4 are well written and easy to follow. The overall construction of Section 5 is too but suffers from perhaps trying to do too much with limited space. As a result there are a lot of important details omitted. The primary detail is that it's unclear what the performance curves are in Figures 2, 3 and 5. Are these the held out test colors? Are these training curves with a disruption of the "test" settings being introduced?

There are no real reasons why the specified domains were selected. Why were the baselines selected? Why were the specific base RL algorithms chosen? Were there any motivating characteristics that led to their selection?

## Quality

Along the lines of the concerns about the clarity of the experiments. It's unclear whether the baselines are properly executed. For example DBC is shown to hardly learn anything in these environments while in the original paper the algorithm does well. An egregious example of this is on the finger/spin and walker/walk domains. Without explanation of how the experimental domains differ from those executed in Zhang, et al. this leads one to believe that the baselines are not fairly compared to the proposed TED auxiliary task. **This is a major concern**.

## Novelty/Significance

As stated above, I fear that the approach is so focused on complex solutions to a relatively simple question (e.g. we want observations from consecutive time points to be closer in representation space, agnostic of distractors) that there hasn't been enough attention placed on more straightforward questions. This also raise some questions about the significance/novelty of the proposed TED module... I wonder if focusing the representation learning to only consider the single next observation as "similar" is a significant drawback as observations within the same trajectory but at a longer temporal horizon may be as equivalently informative when constructing a representation. I wish there was some thought/discussion in the paper about *why* the focus is solely on single-step transition correspondence. As such, I have a hard time really ascribing much significance to the conceptual grounding for the setting of this paper.

As a recommendation/suggestion. It would've been really interesting to see what a TED-augmented version of the baseline comparisons would achieve on these domains. e.g. would CURL+TED improve upon CURL?

## Reproducibility

The concepts surrounding the development of the proposed TED are pretty straightforward that it wouldn't be too difficult to reproduce. However, as described above, the construction of the SSL batch for each transition could stand to be better outlined as well as the overall experimental procedure (e.g. how are the TED losses combined across the different samples for each transition in the batch?).

**Strength And Weaknesses:**

## Strengths

- The paper is well written and is very clear through the motivation and development of the proposed TED auxiliary task. The clarity falters somewhat in the experimental setup and results sections (specifically addressed in the next section). For the most part, I found the paper easy to understand and could follow the conceptual build-up arguing for the use of SSL techniques when considering representation learning for RL.
- I appreciate the depth of discussion about the conceptual framework built through related work. This is a great example of good scholarship on the behalf of the authors. I feel confident about where to place the paper's significance as a result of the detailed overview and technical support derived from previous work. There are some missing references however. In particular, there has been a collection of papers looking into SSL within Deep RL in the recent few years that should be cited (more on this below).
- The construction of TED as an auxiliary task is straightforward and seems like it would be straightforward to incorporate into future research. The major challenge is the engineering aspect of appropriately constructing the batch for SSL. It's promising that the authors intend to release their code to smooth over this barrier.
- The TED augmented algorithms are quite thoroughly compared to relevant baselines on a variety of environments demonstrating the utility of the approach. It was really nice to see the proposed TED applied to different algorithms to demonstrate that the proposed approach is more flexible rather than taking advantage of unique qualities of a single approach/problem domain. The rigor of the experimental analysis is completed with an ablation study showing that the inclusion of the different components in the SSL batch help contribute to the overall performance.

## Weaknesses
- The deviations considered as unrelated/related between episodes are quite narrow. It seems that only color is considered where there are potentially more impactful deviations that are not included (e.g. viewing angle, perturbations in the dynamics, occluded aspects of the scene, etc.). Aside from Figure 1 and a limited collection in the Appendix, there are no visual demonstrations of what the environment variations are. This would be a great inclusion in Section 5.1 to solidify what exactly is meant by distractor and relevant/irrelevant variations. So much of the discussion and explanation of these terms (throughout the whole paper) relies perhaps too much on these specifics being completely understood by the reader.
- The paper doesn't completely outline how $\mathcal{L}_{TED}$ is integrated into the RL loss. It is mentioned that the representation and policy learning are performed hand-in-hand. In algorithm 1 there's no indication of what is done to accumulate the loss with the different samples $x\in\{X', X'', X'''\}$ for every transition in B.
- The clarity of the experimental evaluation is much worse than the preceding sections (specific details in the next section) which casts some doubt on the overall utility of the the approach.
- Figure 1 could probably be better detailed in the text of Section 4.1. There are some missing points of notation (e.g. $z_{\tilde{t}}^{\tilde{e}}$ as well as an unclear differentiation between where the representation of the next observation $z_{t+1}^e$ is used in the RL algorithm. Is it assumed that the RL algorithm is multistep? Or is this an implied use of the next observation within the "target" of a TD-error update?
- The disentanglement metric comes across as an afterthought and is not very clear how and where it's used to evaluate the performance. E.g. how significant is a score of 0 vs. a score of 1? Without this insight, the tables in "Figure" 4 are relatively meaningless...
- There is inconsistency between experiments where the same algorithm is used in the same domain (e.g. CURL/DBC/DrQ on cartpole swingup and panda reach). There is no explanation of where the different experiments differ. Are the results of provided in "Figure" 4 a single run of these algorithms? As far as I can see, the only change between these experiments is the base algorithm that TED was applied to (RAD --> SVEA). Additionally, it seems that the experiments are set-up much differently since there are different time scales (and performances of these baselines) between Figure 2 and Figure 3. *This is a major concern.*
- The ablations are only performed on one algorithm-TED combo. How generalizable are the insights seen in Section 5.3?
- There is a claim that TED enables continual learning without catastrophic forgetting. There are no experiments to test this claim and no other discussion about this term/concept throughout the paper.

### Missing papers

```Mazoure, B., Tachet des Combes, R., Doan, T. L., Bachman, P., & Hjelm, R. D. (2020). Deep reinforcement and infomax learning. Advances in Neural Information Processing Systems, 33, 3686-3698.```

```Schwarzer, M., Rajkumar, N., Noukhovitch, M., Anand, A., Charlin, L., Hjelm, R. D., ... & Courville, A. C. (2021). Pretraining representations for data-efficient reinforcement learning. Advances in Neural Information Processing Systems, 34, 12686-12699.```



**Summary Of The Paper:**

This paper proposes a representation learning method for image-based state observations for RL learning algorithms. This is proposed as an auxiliary task built around self-supervision between successive observations, with a distinct focus on disentangling the learned representations to differentiate between relevant and irrelevant aspects of the environment when learning a policy. The proposed temporal disentanglement (TED) augmentation is then shown to improve performance to a handful of selected RL algorithms.

**Summary Of The Review:**

I enjoyed this paper. I am intrigued by the apparent simplicity of the proposed TED auxiliary task. I am however worried about the limited scope of the variations tested on in the construction and execution of the experiments. I am also fairly concerned about whether the experimental evaluation was properly executed in comparison to baselines across the different domains. There are also concerns about the clarity of how the proposed approach is actually implemented. In light of these concerns I have currently rated this paper as a "borderline reject". I would be interested in raising my score if the authors were able to adequately clarify my questions and some of the concerns raised in my review. I am however not entirely confident that this can be done without substantial revision of the experiments since it's not clear that they were executed fairly.

---

> ### Author Response · Authors · 2022-11-17
> **Response to Reviewer rUnk (1 of 3)**
>
> Thank you for your time and detailed feedback. We are glad you found the writing and motivation clear, and that you appreciate the related work discussion to demonstrate the significance of the paper. We are pleased that you appreciate the concept of using TED, especially the simplicity and flexibility of applying to different base algorithms. We would like to address in detail the specific weaknesses you raised below. We have also uploaded an updated version of the paper with several improvements based on the feedback we received; changes are highlighted in blue for your convenience.
>
>
> ### Weaknesses
>
> > "The deviations considered as unrelated/related between episodes are quite narrow. It seems that only color is considered where there are potentially more impactful deviations that are not included.”
>
> We want to specifically evaluate the ability of learned policies to generalise to unseen values of environment variables when switching from training to test environments. We found that more difficult DMC tasks or more difficult distractors (like camera angle) mean that SOTA algorithms are unable to learn an optimal policy from images in training making it difficult to assess generalisation. Our results show that still many baselines fail to learn an optimal policy in the training environment in some of the tasks. Our results also show that the baselines fail to generalise in our experiments, demonstrating that the colour deviations are already sufficiently difficult to test generalisation performance. We also use Procgen environments in our experiments which have many more factors of variation beyond colour, including factors that affect the optimal policy.
>
>
> > “Aside from Figure 1 and a limited collection in the Appendix, there are no visual demonstrations of what the environment variations are.”
>
> Figures 8 and 9 in the Appendix provide example train and test images for all environments used in the experiments. We have added a reference to these images in the main text of Section 5.1 in the updated version of the paper to help the reader. We also provided a video in the supplementary materials to show a visual demonstration.
>
>
> > “The paper doesn't completely outline how LTED is integrated into the RL loss...”
>
> The exact details of integrating the with RL loss can depend on the implementation of the base algorithm. We require the RL loss and TED loss to be used to update the encoder parameters, which can be done by adding the losses and backpropagating as one loss with a shared optimiser. Our code will be made open-source to provide these implementation details. We also added further details to Appendix B in the updated version of the paper to explain this.
>
>
> > “Figure 1 could probably be better detailed in the text of Section 4.1. There are some missing points of notation (e.g. zt~e~ as well as an unclear differentiation between where the representation of the next observation zt+1e is used in the RL algorithm. Is it assumed that the RL algorithm is multistep? Or is this an implied use of the next observation within the "target" of a TD-error update?”
>
> We have added an explanation of e~ and t~ in the caption of Figure 1 since the notation is specific to the diagram intended to make it clearer where the target encoder and classifier are used to process multiple different types of samples. We use e~ to represent either e or e’ depending on which sample is being processed, and similarly t~ represents either t+1, t’ or t’’ depending on the sample being processed. Regarding the t+1 timestep, this is the next observation received from the environment after the agent performs an action. We assume we have access to a batch of observation and next observation transitions as described in Section 4.1.
>
>
> > “The disentanglement metric comes across as an afterthought and is not very clear how and where it's used to evaluate the performance.”
>
> The disentanglement metric is used to evaluate the extent to which the learned representations have disentangled the factors of variation. We evaluate the degree of disentanglement to demonstrate that TED improves the disentanglement of the base RL algorithm, which we propose as the basis for improving generalisation in this paper. We use an existing metric (Higgins et al. 2017), which is commonly used in the disentanglement supervised learning literature, and we include a small adaptation to allow for frame-stacking. Since it is an adaptation of an existing metric, we introduce it as part of the experimental results analysis in Section 5.1 with an overview to give some intuition behind the results of the metric. Table 1 shows the results of the disentanglement metric.

---

> ### Author Response · Authors · 2022-11-17
> **Response to Reviewer rUnk (2 of 3)**
>
> > “There is inconsistency between experiments where the same algorithm is used in the same domain”
>
>
> We use the original implementations of RAD and SVEA base algorithms to fully demonstrate the flexibility of applying TED without any changes to the base algorithm. This is described in the experimental setup paragraph of Section 5.1, specifically: “following the original setup of the base RL algorithms, the encoder for SVEA has 11 convolutional layers and RAD has 4 convolutional layers. Full implementation details are described in Appendix B.” For a fair comparison to the baselines, we use the same size encoder as the base algorithm in each task as described in the “baselines” paragraph. This explains the difference in performance of the baselines as well as the need to train for more timesteps for the SVEA experiments given the larger number of parameters to be learned. We appreciate the concern if these comments were to be missed, so we have added a sentence to explicitly explain the differences  in the updated version of the paper.
>
>
>
> > “The ablations are only performed on one algorithm-TED combo.”
>
> Due to limits on space and computational resources, we perform these more detailed ablation studies using only one base algorithm, while the main generalisation results are provided with multiple tasks and base algorithms (in Figures 2, 3 and 4).
>
>
> > “There is a claim that TED enables continual learning without catastrophic forgetting.”
>
> This claim is demonstrated through our experiments in Section 5 (Figures 2, 3 and 4), which allows continued learning in the test environment, showing TED improves adaptation. The results demonstrate that TED can equip other algorithms with continual learning without the need to re-learn from scratch after a change in the environment. We have relaxed the claim in the updated version of the paper to use the wording: “...continue learning while reducing catastrophic forgetting”.
>
>
> > “Missing papers”
>
> Thank you for providing these two references, we have added them to Section 2.1 of the updated paper.
>
>
> ### Clarity
>
> > “it's not clear how the stacks of image observations differ between timesteps. Is it that just a single frame is replaced?”
>
> TED does not place any specific requirement on how the frame stacks are created but is flexible based on the base RL algorithm. However, our RAD and SVEA experiments replace only one frame following the setup of the base algorithms. This does mean that much information might be the same but not all this information is likely to be retained in the representation where is it is not required for the value function. The purpose of disentanglement is not to separate the frames, but rather to separate the objects within the stack of frames. For example, separating the trajectory of the cartpole from its colour. Please also note that the PPO experiments do not use frame stacking at all to show the improved generalisation in this different setup.
>
>
> > “The primary detail is that it's unclear what the performance curves are in Figures 2, 3 and 5.”
>
> We kindly refer the reviewer to the opening paragraph of Section 5, where we describe the experiments. The graphs show the average returns of evaluation episodes, the train colours are used up to the vertical dotted line at which point the environment is changed to use the held-out test colours only and we continue training to assess adaptation.
>
>
> > “There are no real reasons why the specified domains were selected. Why were the baselines selected? Why were the specific base RL algorithms chosen?”
>
> We chose the tasks from DMC in which SOTA base algorithms could learn an optimal policy with the colour distractors. We augment this with Panda Reach as more realistic robotics task (that is still achievable with the colour distractors). For the Procgen experiments, we chose tasks that show a clear failure to generalise from the original paper. We kindly refer the reader to the “baselines” paragraph of Section 5.1, where we selected baselines that are current SOTA algorithms covering the different areas of related work we described in the Related Work Section 2.1. The base RL algorithms were chosen as examples of SOTA algorithms for generalisation performance but we hope it is clear the flexibility of TED means it can be applied to other base algorithms as well.

---

> ### Author Response · Authors · 2022-11-17
> **Response to Reviewer rUnk (3 of 3)**
>
> ### Quality
>
> > “It's unclear whether the baselines are properly executed. For example DBC is shown to hardly learn anything in these environments while in the original paper the algorithm does well”
>
> We used DBC's official codebase (found here: https://github.com/facebookresearch/deep_bisim4control). Unfortunately, we could not reproduce the performance shown in the DBC paper with the official codebase. Several GitHub issues show others had similar problems (e.g., https://github.com/facebookresearch/deep_bisim4control/issues/18, https://github.com/facebookresearch/deep_bisim4control/issues/14, https://github.com/facebookresearch/deep_bisim4control/issues/8). The authors suggest trying different hyperparameter and architecture settings in some of these issues. Unfortunately, none of the suggestions fixed the performance. As such, we chose settings for DBC that produced the best results and used those throughout our paper.
>
>
> ### Novelty/significance
>
> > “I fear that the approach is so focused on complex solutions to a relatively simple question (e.g. we want observations from consecutive time points to be closer in representation space, agnostic of distractors)”
>
> The goal of disentanglement is distinct from the other types of representation learning approaches you mention. Requiring observations from consecutive time points be closer together in the representation space does not imply disentanglement and vice versa. It seems reasonable to expect that in a disentangled representation, many dimensions will be similar in consecutive time steps but this is not true in all dimensions and we do not specifically require it. Consider, for example, an object that spawns at a different locations or changes colour within an episode, you could conceivably learn a disentangled representation where consecutive timesteps have a very different values in this dimension and therefore are not close together but are still disentangled. We have extended Section 2.1 with a slightly more detailed comparison to CURL to make sure this point is clear in the paper. Regarding the similarity to approaches that are agnostic to distractors, these aim to ignore distractors without enforcing any structure on the remaining relevant information, whereas TED encourages a disentangled structure to improve generalisation to both variables relevant to the optimal policy as well as distractors as described in Section 2.1.
>
>
> > “I wonder if focusing the representation learning to only consider the single next observation as "similar" is a significant drawback as observations within the same trajectory but at a longer temporal horizon may be as equivalently informative when constructing a representation.”
>
> Whilst we are not using the next observation to explicitly encourage similarity in the representation as described above, we do agree that considering longer time horizons may be an interesting extension of this work, which we have added to Limitations and Future Work Section 6 in the updated version of the paper.
>
>
> > “It would've been really interesting to see what a TED-augmented version of the baseline comparisons would achieve on these domains.”
>
> We show results for the TED auxiliary task applied to three different algorithms in the paper. Where, for example, SVEA would normally be considered as a suitable baseline for RAD-TED. With limited space and computational resources, we are not able to provide results for all baselines with TED applied but hope that the three different base algorithms demonstrate the flexibility of the approach.

---

### Author Response · Authors · 2022-12-05
**Review follow-up**

Dear Reviewers and Area Chair,

We thank you for your reviews and feedback. We are glad that the reviewers all appreciated many aspects of the paper, and we hope that our responses and updates to the paper addressed the concerns that were raised. As we are nearing the end of the discussion phase, please let us know if any of your concerns have not been suitably addressed or if you have anything else you would like to discuss.

Kind regards,
Paper Authors

---

> ### Comment · Reviewer_rUnk · 2022-12-05
> **Re: follow-up**
>
> Thanks for the reminder that we have an active reviewing process on-going. With the preparation leading up to and attending NeurIPS, this slipped my mind.
>
> I was able to re-read my review, the responses the authors provided as well as the revised PDF. I'm happy to say that I've been satisfied by the responses provided by the authors and will indeed increase my score accordingly. I do not have any additional items to discuss as I feel that my concerns were thoroughly addressed in the author response.
>
> Thank you!

---

### Decision · Program_Chairs · 2023-01-20

**Decision:**

Accept: poster

**Justification For Why Not Higher Score:**

Although the paper is well-written and experiments look promising. There is further evidence needed to confirm that the proposed approach is generalizable to other RL tasks. The approach has some limitations (such as the temporal sampling) as pointed out by the reviewers.

**Justification For Why Not Lower Score:**

As mentioned in the decision, I think this paper is a valuable contribution to the ICLR community that would be interested in RL.

**Metareview: Summary, Strengths And Weaknesses:**

### Summary

The paper proposes a method called Temporal Disentanglement (TED) for RL problems where the state is represented in terms of image-based observations. To achieve this the paper proposes an auxiliary loss that does self-supervision between observations based on what is irrelevant and relevant for the task. The experiments focus on Panda Gym and DM Control suite environments and the paper shows improvements over base RL agents.

Below I will summarise some of the  strengths and the weaknesses pointed out by the reviewers:

### Strengths

- The paper is well-written and clear. The TED approach is well-motivated.
- The approach is simple and straightforward. It is relatively easy-to-implement.
- Thorough experiments and comparisons.
- The claims that are being made are sufficiently supported by the experiments.
- The idea of combining the TED auxiliary loss with RL is novel.

### Weaknesses

- The deviations considered as unrelated/related between episodes are narrow.
- The lack of details on how the TED loss is incorporated to RL.
- The temporal sampling can be a limitation of TED.
- In many experiments, the training does not seem to have converged yet.

### Decision
Improving generalization of RL algorithms is an important under-studied area. This paper proposes a novel auxilary loss that seems to be effective in terms of improving the generalization of the RL algorithms. The proposed approach is simple and easy-to-implement. The claims are well-supported by the experimental evidence. The authors did a good job in terms of addressing some of concerns raised by the reviewers. Thus, I am recommending this paper for acceptance.

**Note From Pc:**

if the above contains the word "oral" or "spotlight" please see: "oral" presentation means -> notable-top-5% and "spotlight" means -> notable-top-25%. As stated in our emails, we are disassociating presentation type from AC recommendations